# Characterization of Fruit Development, Antioxidant Capacity, and Potential Vasoprotective Action of Peumo (*Cryptocarya alba*), a Native Fruit of Chile

**DOI:** 10.3390/antiox10121997

**Published:** 2021-12-15

**Authors:** Mónika Valdenegro, Maricarmen Bernales, Marcela Knox, Raúl Vinet, Eduardo Caballero, Aníbal Ayala-Raso, Denisa Kučerová, Rohitesh Kumar, Jitka Viktorová, Tomáš Ruml, Carlos R. Figueroa, Lida Fuentes

**Affiliations:** 1Escuela de Agronomía, Facultad de Ciencias Agronómicas y de los Alimentos, Pontificia Universidad Católica de Valparaíso, Casilla 4-D, Quillota 2260000, Chile; monika.valdenegro@pucv.cl (M.V.); maricarmen.bernales.p@mail.pucv.cl (M.B.); 2Laboratory of Pharmacology, Center of Micro Bioinnovation (CMBi), Faculty of Pharmacy, Universidad de Valparaíso, Valparaíso 2360102, Chile; marcela.knox@uv.cl (M.K.); raul.vinet@uv.cl (R.V.); 3Centro Regional de Estudios en Alimentos Saludables (CREAS), CONICYT-Regional GORE Valparaíso Proyecto R17A10001, Avenida Universidad 330, Placilla, Curauma, Valparaíso 2362696, Chile; ecaballero@creas.cl; 4Instituto de Estadística, Facultad de Ciencias, Universidad de Valparaíso, Valparaíso 2360102, Chile; anibal.ayala@postgrado.uv.cl; 5Department of Biochemistry and Microbiology, University of Chemistry and Technology Prague, Technická 5, 166 28 Prague, Czech Republic; kuceroad@vscht.cz (D.K.); rohiteshkumar@gmail.com (R.K.); Jitka.Prokesova@vscht.cz (J.V.); Tomas.Ruml@vscht.cz (T.R.); 6Laboratory of Plant Molecular Physiology, Institute of Biological Sciences, Universidad de Talca, Talca 3465548, Chile; cfigueroa@utalca.cl

**Keywords:** functional food, antioxidants, polyphenols, bioactivity assessments, anti-inflammatory activity, vascular protection

## Abstract

The peumo (*Cryptocarya alba*) is a native fruit from central Chile that belongs to the Lauraceae family. To characterize the development and the potential health benefits of this edible fruit, quality and physiological parameters, along with antioxidant capacity, were evaluated during three clearly defined developmental stages of the fruit in two seasons. The most distinguishable attributes of ripe fruit were the change in size and color. Low CO_2_ production and no detectable ethylene levels suggested non-climacteric behavior of the peumo fruit. Peumo demonstrate a significant increase in their antioxidant capacity per 1 g of fresh weight (FW) of the sample, from small to ripe fruit. Higher values in ripe fruit (FRAP: 37.1–38.3 µmol FeSO4/gFW, TEAC: 7.9–8.1 mmol TE/gFW, DPPH: 8.4-8.7 IC_50_ μg/mL, and ORAC: = 0.19–0.20 mmol TE/gFW) were observed than those in blueberry fruit (FRAP: 4.95 µmol FeSO4/gFW, TEAC: 1.25 mmol TE/gFW, DPPH: 11.3 IC_50_ μg/mL, and ORAC: 0.032 mmol TE/ gFW). The methanol extracts of ripe fruit displayed the presence of polyphenol acids and quercetin, an ORAC value of 0.637 ± 0.061 mmol TE per g dried weight (DW), and a high cellular antioxidant and anti-inflammatory potential, the latter exceeding the effect of quercetin and indomethacin used as standard molecules. Also, the assay of isolated rat aorta with endothelium-dependent relaxation damage demonstrated that the peumo extract induced vascular protection, depending on its concentration under a high glucose condition. These results demonstrate that these endemic fruits have a good chance as ingredients or foods with functional properties.

## 1. Introduction

The peumo [*Cryptocarya alba* (Molina) Looser] is a Chilean Lauraceae tree with an endemic spread from Maule to the Araucania Regions. It is considered a threatened species in some areas of Chile, mainly due to overexploitation and habitat destruction [1]. Concerning its ecological importance, peumo is one of the representative species of the sclerophyllous forest of the central zone of Chile, including boldo (*Peumus boldus*), quillay (*Quillaja saponaria*), and hawthorn (*Acacia caven*) [2]. All sclerophyllous species are frequently used in low water consumption gardens. Likewise, quillay and boldo are species with interesting pharmacological and industrial applications of their compounds, being an important source of saponins [3] and boldine [4], respectively. 

Despite no agro-industrial use, peumo leaves have been used in traditional medicine like infusion or ointment [2,5]. On the other hand, this tree has beautiful and pink-colored berries (Figure 1), called peumos, collected and consumed by the Mapuche Amerindians, principally as a cold infusion, since pre-Colombian times. This fruit is composed of edible and pink skin, with intense flavor and aroma at maturity, and a large seed like a nut; these characteristics have allowed its gastronomic use in recent years [2].

The essential oil of this species was reported to be composed mainly of 1-terpinen-4-ol and p-cimol [6], while the cryptofolione derivative has been the only compound isolated from the edible fruits [7]. Domínguez and Martínez (2002) [8] reported that the peumo fruit skin has many polyphenols, but it is unclear if this potential is linked to a particular genotype. Simirgiotis (2013) [9] reported a high antioxidant capacity (9.12 ± 0.01 mg·mL^−1^), determined by DPPH assay in ripe fruit and flavonoid glycosides, phenolic acids, anthocyanins, and flavonoid aglycons as the major phenolic compounds in fruit and aerial parts of peumo extracts.

The antioxidant capacity of native Chilean fruits like murtilla (*Ugni molinae* Turcz.) (10,770 ± 453), maqui (*Aristotelia chilensis* (Molina) Stuntz) (19,850 ± 966), and calafate (*Berberis* sp.) (25,662 ± 3322), determined by oxygen radical absorbance capacity (ORAC: μmol TE/100 g fresh weight), has been described as higher than commercial berries such as raspberries (6903 ± 1019), blueberries (8869 ± 334), and blackberries (9043 ± 1253) [10]. Furthermore, this antioxidant capacity has been associated with a high functional potential [10,11,12]. Likewise, water soluble extracts of maqui berry have been reported to prevent the oxidation of copper-induced low-density lipoprotein (LDL), adipogenesis, and inflammatory actions, and to protect human endothelial cell cultures [12,13,14,15,16].

The quality, physiological parameters, and potential health benefits of many attractive native fruits could be affected by inadequate handling postharvest. Therefore, knowledge about the fruit’s physiological and physicochemical parameters before studying the healthy potential of native fruit is relevant. To the best of our knowledge, no characterization of fruit ripening progression has been carried out on the previous described native Lauraceae of Chile. Therefore, it seems interesting to study the relationship between maturation and bioactive compounds in peumo fruit. The present research aimed to describe the development of peumo (*C. alba*) fruits, evaluate the change of antioxidant capacity and polyphenols during fruit development, and examine the ability of ripe fruit extracts to act as a vascular defender.

## 2. Materials and Methods

### 2.1. Plant Material

Fruits of peumo (*C. alba*) were picked from an ornamental population located at La Palma, Valparaiso Region, Chile (32°52′48″ S 71°14′51″ W, 462 masl) during two times: April 2017 and 2018. Fruits from both periods without peel injuries were classified into three distinct developmental stages according to shape, weight, and color modifications: Ca1, small and slim clear pink fruit; Ca2, medium and slim pink fruit; Ca3 dark pink fruit (Figure 1). 

### 2.2. Fruit Quality Assessments

Fifty fruits from each stage and time were weighed, determined, and analyzed for humidity according to AOAC protocols [17]. The water activity was determined using a 4T AQUALab (Decagon Devices, Inc.; Pullman, WA, USA). Fruit firmness was measured using a compression tester (Brookfield CT3, AMETEK; Middleboro, MA, USA) using a TA4/1000 probe and a plate probe of 10 cm in diameter, at a speed of 0.5 mms−1 for 2 s. One the compression cycle was applied, the texture profile parameter (hardness) was defined as the maximum force required to compress the sample, calculated as the peak force (N) of the first compression of the product. Two measurements were performed on 25 fruits (each equatorial side) per stage and registered as Newtons (N). The color of the peel was determined in 25 fruits (measured on each equatorial zone) per developmental stage and recorded using a chroma meter (model CR-300, Konica Minolta; Tokyo, Japan) coupled to a Minolta DP-301 data processor. The measure was detailed as the CIELAB scale (L*, a*, b*, hue angle-h°, and chroma-C). After color recordings were performed, the fruits were frozen in liquid nitrogen and stored at −80 °C until use.

The analysis of pH, titratable acidity (TA), and soluble solids content (SSC) were performed in four replicates for each fruit stage, in each season. Six grams of tissue from each sample replicate was powdered, filtered through Miracloth, and evaluated as follows: SSC was determined at 20 °C using a handheld temperature compensated refractometer (Atago, Tokyo, Japan), and recorded as °Brix; the pH of the juice was recorded with a pH-meter (model PL-500, Ezdo; Taipei, Taiwan); and TA was carried out by titration of an aliquot of 10 mL of diluted fruit juice with 0.1 mM NaOH, and registered as a % of organic acid. All chemical reagents were purchased from Merck Company (Darmstadt, Germany).

### 2.3. Ethylene Production and CO_2_ Production

The ethylene production was determined using four replicates of 35 g of fresh fruit (for each stage from each seasons). Each sample was introduced into tightly closed chambers (400 mL) and maintained for 2 h at 20 °C. Gaseous samples of one milliliter each were withdrawn from the headspace volume of the chambers and measured for ethylene in a gas chromatograph (model 7820A, Agilent Technologies; USA), equipped with a flame ionization detector and an Elite Plot Q column (30 m × 0.32 mm i.d.) at 80 °C, utilizing helium as the carrier gas (50.3 cm/s). The injector and detector temperatures were 120 °C and 155 °C, respectively, and ethylene samples of known concentration (100 µL/L) were regularly used for equipment calibration. Three independent ethylene samples were taken per chamber, and the results were expressed as nL ethylene/kg FW h. The CO_2_ concentrations were recorded in the chambers, as mentioned above, using a CO_2_ detector (MAP Headspace Gas Analyzer, Checkpoint II, Dansensor; Brooklyn Park, MN, USA) and registered as mg CO_2_/kg FW h.

### 2.4. Determination of Antioxidant Capacity, Total Polyphenol Content, and Total Flavonoid Content during Fruit Development

Four different procedures for determining the antioxidant capacity were used: the ferric reducing antioxidant power (FRAP), the Trolox equivalent antioxidant capacity (TEAC), 2,2-diphenyl-1-picrylhydrazyl (DPPH) assay, and Oxygen Radical Absorbance Capacity (ORAC). For each assay, four replicates of fruit tissue for each stage and season were used. Two grams of the sample was powdered in liquid nitrogen, mixed with 12 mL of 80% methanol, extracted at 200 rpm for 1 h, and filtered through a 0.45 µm micropore membrane for all methods. Besides, four replicates of ripe blueberry (*Vaccinium corymbosum* ‘Bluegold’) fruit were collected from an orchard located in La Palma and utilized as reference fruit for the analyses above, during the 2018 season. All chemical reagents were purchased from Merck Company (Darmstadt, Germany).

#### 2.4.1. The Ferric Reducing Antioxidant Power (FRAP) Assay

The FRAP was performed according to Benzie and Strain (1996) [18]. Therefore, 3 mL of fresh FRAP reagent was mixed with 100 μL of methanol extract previously diluted in 300 μL of distilled water. The samples were incubated at room temperature for 5 min. The OD was recorded at 593 nm in a UV/Vis spectrophotometer (model Lambda 25, Perkin Elmer; Akron, OH, USA), and the results were expressed as µmol FeSO4/g FW. The regression equation of the calibration curve was y = 0.5908x + 0.0053 with a R^2^ = 0.9989. 

#### 2.4.2. The Trolox Equivalent Antioxidant Capacity (TEAC) Assay

The standard TEAC assay was carried out according to Van den Berg et al. (1999) [19] with modifications [20], using Trolox as the standard. The methanol extract was diluted in a phosphate-buffered saline (PBS) solution (pH 7.4), and 40 μL was blended with 1.96 mL of the radical solution. The decrease in the OD at 734 nm in a UV/Vis spectrophotometer (model Lambda 25, Perkin Elmer, Akron, OH, USA) was measured for 6 min and used to calculate TEAC according to Murcia et al. (2009) [21]. The regression equation of calibration curve was y = 3.6085x + 10.34 with an R^2^ = 0.9976. ABTS+ scavenging effect (%) = 1 − (AS/AC)) × 100, where AC is the absorbance of uninhibited radical cation (ABTS^•^ + alone) and AS is the absorbance, measured 30 min after the addition of antioxidant test samples. The results were determined as mmol Trolox equivalents (TE)/g FW. 

#### 2.4.3. The 2,2-diphenyl-1-picrylhydrazyl (DPPH) Assay

The free radical scavenging activity was measured using Trolox as the standard and by the DPPH assay [22]. Briefly, 100 µL of methanol-extract dilution was blended with 3.9 mL of 103.5 µM solution of the DPPH radical in methanol and incubated at room temperature for 60 min in the dark. The reduction of the DPPH radical was analyzed by the Optical Density (OD) measurement at 517 nm in a UV/Vis spectrophotometer (model Lambda 25, Perkin Elmer, Akron, OH, USA). The regression equation of the calibration curve was y = 9.7517x − 0.0308 with an R^2^ = 0.9922. The absorbance change (%) = [(AC − AS)/(AC − AP) × 100], where AC is the absorbance of the control, AS is the absorbance of the extracted sample, and AP is the absorbance of the positive control. Then, IC_50_ [μg/mL] was determined as the extract concentration to 50% of the DPPH scavenging capacity.

#### 2.4.4. The Oxygen Radical Absorbance Capacity (ORAC) Assay

The ORAC activity was evaluated as reported by [10], using 2,2′-azobis(2-amidinopropane) dihydrochloride (AAPH) as a source of peroxyl radicals and fluorescein as an oxidizable probe. In brief, 45 μL of the methanol extract (diluted in 75 mM phosphate buffer, pH 7.4) was added to 96-well microplates, each containing 75 μL of APPH (18 mM) and 200 μL of fluorescein (108 nM). The plates were placed in a Multi-Mode Microplate Reader (Fluoroskan Ascent, Thermo Scientific; Waltham, MA, USA) and incubated at 37 °C for 60 min, with shaking every 3 min. The fluorescence (485 nm Ex/538 nm Em) was monitored every 3 min throughout the experiment. The analysis of each sample was performed in triplicate. The ORAC activity results were estimated based on a standard curve of Trolox using a quadratic regression equation obtained between the Trolox concentration and the net area under the fluorescence decay curve. ORAC activity was expressed as mmol of TE/g FW. The regression equation of calibration curve was y = 0.2406x + 4.0453 with an R^2^ = 0.9641.

#### 2.4.5. Total Phenolic Content Determination

Total phenolic content (TPC) was determined according to Singleton and Rossi (1965) [23] and Galati et al. (2003) [24]. First, one hundred microliters of methanol-extract dilution were mixed with 50 µL Folin-Ciocalteu reagent and incubated for 5 min at room temperature. Then, 150 µL of 20% (*w*/*v*) Na_2_CO_3_ and 700 µL distilled water was added, and the mixture was incubated at room temperature in darkness for 30 min. At the end of the incubation period, 4.5 mL distilled water was added to 500 µL of the mixture. The OD was measured at 760 nm in a UV/Vis spectrophotometer (model Lambda 25, Perkin Elmer, Akron, OH, USA), using gallic acid (GA) as the standard. The results were expressed as mg gallic acid equivalent (GAE)/g FW. The regression equation of calibration curve was y = 4.6334x + 0.036 with an R^2^ = 0.987. 

#### 2.4.6. Total Flavonoid Content Determination

Total flavonoid content (TFC) was analyzed according to Chang et al. (2002) [25]. Five hundred microliters of methanol-extract dilution were mixed with 1.5 mL of 95% ethanol, 100 µL of 10% AlCl_3_, 100 µL of 1 M CH_3_CO_2_K, and 2.8 mL of distilled water. After incubation for 30 min at room temperature, the OD of the reaction mixture was determined at 415 nm in a UV/Vis spectrophotometer (model UV-160A, Shimadzu; Kyoto, Japan), using quercetin-3-glucoside as the standard. The results were expressed as mg quercetin equivalent (QE)/g FW. The regression equation of the calibration curve was y = 5.7749x with an R^2^ = 0.9862. 

### 2.5. Bioactivity Assay of Ripe Peumo Extract

#### 2.5.1. Functional Extract Preparation

Fruit samples of the ripe stage (Ca3) from the 2018 period were cut pieces. Then samples were oven-dried at 60 °C for 48 h and finely pulverized using a windmill MF 106 (IKA, Nara, Japan). The plant powder (sieve 0.3–0.5 mm particle size) was kept in sealed sterile containers at −20 °C (Electrolux; Stockholm, Sweden) until solvent extraction [26]. Four independent extractions of ten grams of samples were blended in an orbital shaker (Cole-Parmer; Vernon Hills, IL, USA) at 100 rpm with 150 mL of 100% methanol at room temperature for 24 h [26,27]. After extraction, each sample was filtered using a Miracloth and then through a Whatman No. 1 filter (Merck, Darmstadt, Germany) to obtain a clear extract. Filtered extracts were then dried at 40 °C under reduced pressure using a rotary evaporator (Büchi, Labortechnik AG; Switzerland). The concentrated extracts were reconstituted with methanol to a concentration of 500 mg/mL. Extracts were filter-sterilized through a 0.45 µm syringe filter (Merck, Darmstadt, Germany) and stored in sterile containers at −20 °C until U-HPLC/MS analysis, TPC, TFC, and ORAC determinations, described above, and bioactive assays. 

#### 2.5.2. Ultrahigh-Pressure Liquid Chromatography-Mass Spectrometry (U-HPLC/MS) Analysis

The U-HPLC/MS analysis was performed by the Laboratory of Mass Spectrometry at the Central Laboratory of UCT, according to Viktorová et al., 2020 [27]. The determinations were performed on a Luna C18 Phenomenex 150 × 2 mm, 3.1 µm column on Thermo LC-MS LTQ—Orbitrap Velos spectrometer (Thermo Fisher Scientific; Waltham, MA, USA) in both negative and positive modes. A gradient of 100% water (+0.1% formic acid) to 100% methanol (+0.1% formic acid) was used as the mobile phase over a time of 15 min. The data obtained were analyzed using Xcalibur 2.2 (Thermo Fisher Scientific, Waltham, MA, USA). Molecular ion adducts such as [M + H]^+^, [M + Na]^+^, [2M + H]^+^, [2M + Na]^+^, [M-H]^−^, [2M-H]^−^, and [2M-H + Na]^−^ were manually identified in order to determine the molecular ion [M]. Once the major molecular ions were identified, a molecular formula (MF) was generated using Xcalibur, and a search using the [M] and MF filter was performed on Scifinder (https://scifinder-n.cas.org/, accessed on 15 July 2021) to identify tentative compounds of ripe fruit extract. The compounds were detected on the agreement of recorded MS/MS spectra with online mass spectral libraries (such as ‘METLIN’ and ‘mzCloud’) or the scientific literature. For some of the detected compounds, several chromatographic peaks meeting the HRMS criteria were observed, likely indicating the presence of structural isomers.

#### 2.5.3. Cellular Antioxidant Activity

Cellular antioxidant activity was measured according to [28]. Briefly, macrophages (Raw 264.7, Merck; Kenilworth, NJ, USA) were cultivated in a complete medium -Dulbecco’s modified eagle’s medium, DMEM (Merck, Darmstadt, Germany), according to information of the manufacturer and supplemented with 10% Fetal Bovine Serum (FBS, Merck, Darmstadt, Germany) and Antibiotic Antimycotic Solution (Merck, Darmstadt, Germany). When the cells reached 80% confluence, they were passaged by scraping and dilution. Cells between 5–20 passages were used for the experiment. The cells were seeded at a 1 × 106/mL density in DMEM medium. After 24 h, the cells were washed with PBS and DCFH-DA (2′,7′-Dichlorodihydrofluorescein diacetate, Merck, 0.0125 mg/mL in medium without FBS) was added. Then, the sample extracts were added to the final concentrations of 15–500 mg/L. After 1 h of co-incubation, the medium was replaced with 44 µM 2,2’-Azobis (2-methylpropionamidine) dihydrochloride (AAPH; VWR; Radnor, PA, USA) in PBS and fluorescence (ex./em. 485/540 nm) was immediately recorded at 5 min intervals for 1 h using the SpectraMax i3x Multi-Mode Detection Platform (Molecular Devices; Silicon Valley, CA, USA). The experiment was done in three biological repetitions. Quercetin (Merck, Darmstadt, Germany), in a concentration range 0.6–20 µM, was used as a positive control. 

#### 2.5.4. Anti-Inflammatory Activity

Anti-inflammatory activity was measured according to [29]. Briefly, macrophages were seeded as above described. After 48 h, the cells were washed with PBS. The MEM medium enriched with 100 ng/mL lipopolysaccharide (LPS; Merck, Darmstadt, Germany) was added to the wells, together with a sample concentration range of 0.01–2.5 g/L. A non-steroidal anti-inflammatory drug, indomethacin (Merck, Darmstadt, Germany), was tested as a positive control in the concentration range 6.25–100 µM. After 24 h, the medium was used to quantify nitric oxide (NO) and inflammatory cytokines. NO production was quantified by 0.04 g/mL Griess reagent (Merck, Darmstadt, Germany), prepared in deionized water, after 15 min incubation and absorbance measurement at 540 nm. The cell viability was checked by resazurin (Merck, Darmstadt, Germany) −0.03 mg/mL in PBS- assay, where the fluorescence was recorded (560/590 nm, ex./em.). Both TNF-α and IL-6 were determined by uncoated ELISA, performed according to manufacturer’s instructions (Interleukin-6 Mouse uncoated ELISA kit (Invitrogen, Thermo Fisher Scientific; Waltham, MA, USA), Mouse TNF alpha uncoated ELISA kit (Invitrogen, Thermo Fisher Scientific; Waltham, MA, USA)). IC_50_ was calculated using the Quest Graph IC50 Calculator (AAT Bioquest; Sunnyvale, CA, USA) and values of relative activity were expressed as the ratio of (sample–untreated cells) and (LPS-stimulated cells–untreated cells).

#### 2.5.5. Animals

Male Sprague-Dawley rats (weighing 240–280 g) housed under standard environmental conditions with free access to food and water were used. All procedures were approved by the Ethics Committee of the Universidad de Valparaíso (CIBICA 065-2015) and also complied with the Guide for the Care and Use of Laboratory Animals (National Research Council, 2011) [30]. All chemical was from Merck Company (Darmstadt, Germany).

#### 2.5.6. Rat Aortic Rings Preparation and Recording

Rats were sacrificed by cervical dislocation after CO_2_ inhalation, and the thoracic aorta was carefully removed and mounted on a tissue chamber as previously described [31]. Briefly, the aorta was carefully dissected, cleaned of connective tissue, and divided into three-ring segments of 5 mm. The most outstanding care was taken without touching the vessel’s lumen to preserve the endothelium’s integrity and functionality. The aortic rings were placed between two stainless steel hooks and immersed in a 20 mL organ chamber containing a modified Krebs-Henseleit buffer (KHB) with 5 mM D-glucose and 20 mM mannitol (pH 7.4). The KHB was kept at 37 °C and was constantly aerated with a 95% O_2_–5% CO_2_ gas mixture. The isometric tension of the vessels was recorded using force transducers integrated into the Myobath II system (WPI; Sarasota, FL, USA). The rings were equilibrated for 60 min under an optimal baseline tension of 1.5 g. The mechanical stability of the system was achieved by adding a depolarizing solution of KHB (70 mM KCl) until reaching a maximum stable contraction of the aortic rings.

#### 2.5.7. Evaluation of the Protective Effect of the Ripe Peumo Extracts

The potential vasoprotective effect of the extract was measured by its ability to prevent endothelium-dependent relaxation damage in vessels acutely pre-incubated (3 h) with high glucose (HG, 25 mM D-glucose), as previously described [32]. First, endothelium functionality was checked by testing the ability of the 0.1 µM phenylephrine (PE) precontracted vessel to relax against a 1 µM acetylcholine (ACh) stimulus. Subsequently, three rings obtained from the same aorta were prepared and incubated independently, for 3 h, under the following conditions: KHB (Control), KHB with HG (25 mM D-glucose), and KHB with HG plus the peumo extracts (0.1, 1, and 10 mg/mL). After incubation, vessels were precontracted with 0.1 µM PE, and once stable contraction was achieved, concentration-response curves for ACh were obtained. The maximum relaxation (Emax) and the concentration that produces 50% of the maximum effect (EC50) for ACh were estimated from the records obtained. EC50 values were obtained by nonlinear regression with a four-parameter equation using GraphPad Prism version 6 (GraphPad Software; La Jolla, CA, USA).

### 2.6. Statistical Analysis

The experiments were conducted using a completely random design with four biological replicates using three technical replicates. All results were expressed as the mean ± standard error (S.E.). The data were analyzed by one-way ANOVA; a principal component analysis (PCA) and correlation matrix were performed using R Statistical Software (R Core Team, 2013; R Foundation for Statistical Computing, Vienna, Austria) [33]. 

## 3. Results

### 3.1. Characterization of Quality and Physiological Parameters during Fruit Development of the Peumo Fruit

The present study classified three developmental stages of peumo (*C. alba*) fruits (Figure 1). A constant boost in fruit fresh and dry weights was registered according to ripening during the first harvest season, and non-significant differences between Ca2 and Ca3 stages were observed during the second season (Table 1). The water activity displays non-significant differences during ripening, with a similar decrease throughout both seasons. The fruit length increased during both seasons, while the diameter only increased for the 2017 season; despite the above, the fruits have a thin shape in all analyzed stages and both seasons (Figure 1, Table 1). The fruit firmness displays a constant reduction during ripening in both seasons, with firmness in ripe fruit near 5 N (Table 1).

The pH range during the development of peumo fruits was 5.6–6.3. Titratable acidity (TA) could not be determined, as no drastic changes were detected in the titration curve. A significant soluble solids content (SSC) increase was observed only for the 2017 season from the Ca1 to Ca2 stages, with an SSC of 24–38° Brix in ripe peumo fruit (Table 1).

Concerning color changes during fruit development of the peumo (Table 1), the hue angle (h°) dislayed a significant decrease during ripening in both seasons. In contrast, the L* and b* decreased significantly only in the second season. A higher increase of a* in the ripe stage was observed in the 2018 season. Nevertheless, the instrumental color obtained suggests the increase in red color during peumo ripening; the obtained data was not precise according to the visual color (Figure 1). 

In the present study, the CO_2_ production of fruits decreases continually until the end of ripening during both seasons (Table 1). Indeed, ethylene production was not detected at any stage of peumo fruits (Table 1) in both seasons, suggesting a fruit’s non-climacteric behavior.

### 3.2. Antioxidant Capacity, Total Polyphenol, and Flavonoid Content during Fruit Development of the Peumo Fruit

Determinations of total antioxidant capacity by four different methods (FRAP, TEAC, DPPH, and ORAC assays) indicated that the increase in antioxidant capacity, according to the ripening progress, displayed significant differences between the Ca1 and Ca3 stages. This trend was similar to those observed for total polyphenols content (TPC) and total flavonoids content (TFC) (Table 2). However, no differences were observed in the ORAC method for the second season. The TPC in ripe peumo was near 17 mg GA/g FW, and TFC was near 9 mg QE/g FW in both seasons; these values were higher than those determined for ripe blueberries in the 2018 season (TPC:2.75 mg GA/g FW and TFC: 2.12 mg QE/g FW). 

The principal component analysis (PCA) of antioxidant capacity for the 2017 (Figure A1A) and 2018 (Figure A1B) harvest seasons describes a similar behavior for FRAP, TFC, and TEAC analysis. However, DPPH and TPC have different behavior for each season, with a high correlation with the other antioxidant variables in the 2017 season and a mainly orthogonal location (not correlated) for 2018 season. The correlation between antioxidant analysis for the 2017 (Table A1) harvest season describes a significant correlation, where FRAP, TEAC, and TFC are the most correlated. Nevertheless, we have a different correlation for the 2018 (Table A2) season, where TPC and DPPH describe not correlated behavior. However, FRAP, TEAC, and TFC still present a positive correlation between them in the season.

### 3.3. Composition of Peumo Fruit Extract

The chemical composition of peumo fruit extract was determined by non-target analysis using U-HPLC/MS LTQ in both positive and negative modes (Figure 2). Identified compounds (Table 3) included many flavonoids, alkaloids, and lignins. The main part of flavonoids was represented by quercetin and its derivatives and metabolites, followed by proanthocyanidins (namely procyanidins), phenols (catechin and epicatechin) and polyphenols (chlorogenic acid and its analogue, 4-caffeoylquinic acid), flavones (luteolin 7-O-glucuronide, sexangularetin), and other flavonoids. Lignans were represented by 4-O-methylcedrusin and (+)-lariciresinol. Alkaloids were represented by cryprochine and its stereoisomer. The high content of flavonoids could be responsible for the antioxidant and anti-inflammatory activity of peumo extract. Also, the ORAC value of peumo extract (500 mg/mL) was 0.637 ± 0.061 mmol/g DW (Table 3).

### 3.4. Bioactivity of Peumo Fruit Extracts

The antioxidant capacity of peumo extract in the macrophages was also confirmed (Table 4). Biologically active components of peumo extract were able to enter the macrophages and reduce the oxygen radicals generated by AAPH in a dose-dependent manner. The concentration of peumo extract reducing the radicals to half (IC_50_) was calculated. Quercetin, a nature flavonoid with known strong antioxidant potential, was used for comparison. Our data demonstrated that the peumo extract was only 2.3 ± 0.0 times less active than pure quercetin. This result indicates a strong antioxidant potential of peumo extract.

The anti-inflammatory activity of peumo extract was evaluated as the ability to reduce the production of inflammatory markers in bacterial lipopolysaccharide-stimulated macrophages. The inflammatory markers evaluated were nitrite oxide (NO) and the cytokines TNF-α (tumor necrosis factor) and IL-6 (interleukin). As displayed in Figure 3, peumo extract could inhibit the production of cytokines in a dose-dependent manner. 

Comparing the anti-inflammatory potential of indomethacin and peumo extract (Table 5), we can conclude that peumo is at least 1.5 ± 0.1 times more active than the non-steroidal anti-inflammatory drug indomethacin. 

The anti-inflammatory and antioxidant PCA’s for a peumo lyophilized sample both present an expected behavior. From this analysis, we have an insight that peumo and quercetin behave similarly, with a highly negative correlation with antioxidant capacity analyses (TFC and ORAC); however, TPC was not related with the other variables (orthogonal) when we expected a positive correlation with TFC and ORAC. On the other hand, the anti-inflammatory effect variables behave opposite between IL-6, TNF-α, and NO.

The deterioration of endothelium-dependent relaxation, induced by high glucose in the aorta rings, was partially reversed by the extract of ripe peumo fruits (Figure 4). The protective effect of the extract was dependent on the dose, demonstrating a significant difference for the concentrations of 1 mg/mL (*p* < 0.01) and 10 mg/mL (*p* < 0.001). The ACh pD_2_ value did not display significant differences at any of the studied concentrations of the extract.

## 4. Discussion

The knowledge of physiological and physicochemical parameters is necessary to use traditional food understudied in the food industry [12,26]. The present study demonstrated a constant increase in weight of peumo fruit, thin shape in all analyzed stages, and a continuous reduction of firmness during fruit development, with firmness in ripe fruit near 5 N (Table 1). In Chile, the peumo fruit has been compared with German Peumo (*Crataegus monogyna*) [9]. However, the species of the genus *Crataegus* (hawthorn) display a higher firmness (28.89 to 40.03 N) [34] compared to results observed in peumo.

The development stages of peumo fruit displayed a pH range (5.6–6.3) similar to that reported in bean (pH: 5.6–6.5) and higher than that of the yellow banana (pH: 5.00–5.29), according to the U.S. FDA and the Center for Food Safety and Applied Nutrition (2017) [35]. However, TA could not be determined, since no drastic changes were detected in the titration curve, and a significant SSC increase was observed only for the 2017 season from the Ca1 to Ca2 stages, with an SSC of 24–38 °Brix in ripe peumo fruit (Table 1). As indicated by Vogel et al. (2008) [5], the fruits of peumo contain 70% carbohydrates, 16% crude lipids, 5.6% fiber, and 6% proteins, indicating that the major component of SSC is carbohydrates. Regarding color changes during fruit development, the instrumental color obtained suggests the increase in red color during peumo development; however, the obtained data was not precise according to the visual color (Figure 1).

In the present study, the CO_2_ and O_2_ production and no ethylene production suggested a non-climacteric behavior. Preliminary studies with applications of exogenous ethylene were made (0.1–100 µL L^−1^) on ripe fruit in Ca3. These results did not alter the firmness, color, and respiration rate (not shown or included). Non-climacteric behavior during storage has been reported in fruit with a big seed, such as olive [36]. However, ethylene treatment (0.1–1000 μL L^−1^) of dark green ‘Konservolia’ olives harvested before the green maturation and then storage at 20 °C demonstrated fruit firmness increases in an ethylene concentration-dependent manner [37]. Therefore, further analysis should be performed to understand the hormonal regulation of ripening and the effect of ethylene in the quality parameter of peumo fruit during postharvest, for the adequate storage of this fruit during its commercialization or industrial use.

Similar to that observed in other Chilean native fruits, such as arrayan [26], the present study demonstrated that the total antioxidant capacity, total polyphenols content (TPC), and total flavonoids content (TFC) of peumo increase according to the development progress (Table 2). In the present study, the TPC and TFC in ripe peumo were over 17 mg GA/g FW and 9 mg QE/g FW, respectively, in both seasons. Similar results were reported by Simirgiotis et al. (2013a) [9], with a TPC of 17.70 ± 0.02 mg GA/g FW and 8.22 ± 0.04 mg QE/g FW for peumo ripe fruit, collected in the Biobio Region of Chile (37°08′01.08″ S; 72°43′57.88″ W; 177 masl). Previous studies also indicate that the ripe fruit of Chilean species, such as maqui, murta, and calafate have higher antioxidant capacity (determined by ORAC) and TPC than commercial blueberry [10]. In the present study, the TPC and TFC observed in ripe peumo were higher than those determined for ripe blueberries (2.75 mg GA/g FW and 2.12 mg QE/g FW), suggesting a functional potential for peumo fruit.

The principal component analysis (PCA) of antioxidant capacity for the 2017 (Figure A1A) and 2018 (Figure A1B) harvest seasons, describes a similar behavior for FRAP, TFC, and TEAC analysis. However, DPPH and TPC have dissimilar behavior between seasons. Possibly, for TPC, this could be technical interference from other sugar molecules less present in the 2017 season. Therefore, it has been described that the environmental condition could be related to the differences in sugar contents of native fruits such as arrayan [38]. Otherwise, DPPH behavior may be due to a technical error between season analysis. Nevertheless, in the 2017 harvest season, all describe a significant correlation, where FRAP, TEAC, and TFC are the most correlated (Table A1). The analysis for the 2018 season displays a different correlation pattern, where TPC and DPPH describe a not correlated behavior (Table A2). However, FRAP, TEAC, and TFC still present positive correlation between them in the season. Similarly, good correlations between TPC and antioxidant capacity was described for TPC with FRAP and TEAC during arrayan ripening, and a variable correlation value for TPC and DPPH between harvest seasons of arrayan fruit [26]. 

The short seasonality of the many native fruits makes it difficult to have these fresh fruits for consumption or use in the agroindustry all year, or away from collection sites. Therefore, these products’ uniformity, effectiveness, and richness depend on preserving bioactive compounds throughout the value-added chain [12]. In this sense, we evaluated the main compounds and vasoprotective activity of methanolic extract from oven-dried peumo fruit. This extract of peumo was in a beginning analysis by HPLC-DAD, using commercial standards, displaying catechins, chlorogenic acid and derivates, and quercetins as the major phenolic compounds (data not displayed). However, this technique did not identify several compounds, and a U-HPLC/MS was carried out to identify a more significant number of molecules. In the present study, the main part of flavonoids was represented by quercetin and its derivatives and metabolites, followed by proanthocyanidins (namely procyanidins), phenols (catechin and epicatechin) and polyphenols (chlorogenic acid and its analogue, 4-caffeoylquinic acid), flavones (luteolin 7-O-glucuronide, sexangularetin), and other flavonoids. Flavonoids mainly exist in the form of sugar-conjugated derivatives in nature [39]. Lignans were represented by 4-O-methylcedrusin and (+)-lariciresinol. Alkaloids were represented by cryprochine and its stereoisomer. Mainly, the 3-O-conjugates were detected (Table 3), which are in vivo more bioavailable than quercetin in its aglycone form [40]. Although data on the biological effect of in vitro glycosylation of polyphenols is somewhat inconsistent [41], sugar modification affects their solubility and binding to components in the cultivation medium, mainly serum albumin, which may alter the cellular biology uptake of flavonoids. Our previous U-HPLC/MS analysisdemonstrated that the extract of freeze-dried ripe fruit is rich in polyphenolics, such as chlorogenic acid, and its analogous, procyanidins, catechins, and quercetins, like other reports of extract from peumo fruit [9,42], and the presence of (+)-lariciresinol, a phenylpropanoid compound [27], and some unidentified molecules in the methanolic extract from freeze-dried ripe fruits. 

The high content of flavonoids, such as quercetin, present in Chilean native fruits associated with functional results, suggests its protective action in inflammatory diseases [12]. Purified quercetin has a variety of biological effects, including antioxidant, anti-inflammatory, antiallergic, and platelet antiaggregant effects [43], and potential protective effects against acute lung injury (ALI) induced by Gram-negative bacteria [44]. It is essential to note that the vascular protection of native fruit extracts could be associated with a combination of diverse molecules rather than a particular molecule. 

Different studies have reported the functional properties of phenols (catechin and epicatechin) and polyphenols (chlorogenic acid and its analogue, 4-caffeoylquinic acid), biactive molecules also detected in the present study. For instance, total polyphenols of Erigerontis herba (*Erigeron breviscapus*) displayed vasodilative effects in endothelium-removed aortas through the nitric oxide synthase (NOS) pathway [45]. In turn, extracts of *Crataegus pentagyna* leaf, flower, and fruit, containing high levels of chlorogenic acid and epicatechin, have potential benefits in cardiovascular diseases associated with endothelial dysfunction [46]. Also, polyphenols extracts of crop plants display interesting effects on vasorelaxation. In this sense, hydroxytyrosol from olive oil improved acetylcholine-induced vasorelaxation under oxidative stress conditions [47]. The skin and seeds of grape (*Vitis vinifera*), with high levels of polyphenols (e.g., catechin and glycosylated flavonoids), contain vasorelaxant agents capable of relaxing in vitro vascular preparations in an endothelium-dependent manner [48]. Several reports have demonstrated the beneficial activity and preventive effects of catechins in the prevention of cardiovascular disease (CVD), by regulating lipid metabolism, vascular endothelial protection, and reducing blood pressure [49,50]. However, certain caution has been observed in green tea consumption, as a good source of catechins, to reduce CVD risk due to the safety and other metabolites present in green tea [51]. Cholorogenic acid, a polyphenol acid found in peumo extract in the current research, has been associated with the modulation of lipid metabolism and glucose and vasodilatory action, potentially treating CVD. However, some difficulties in medicinal research have to be countered [52,53,54]. In the present study, the anti-inflammatory effect of peumo extract is more active than the non-steroidal anti-inflammatory drug indomethacin, with significant differences in NO production. However, major studies are necessary to determine the effect of peumo extract on NO production associated with the vascular endothelium. 

The present work in peumo fruit, and previous work using arrayan [26], find that extracts obtained from dehydrated fruit by oven-drying demonstrate that this dehydration technique allows for the preservation of the vasoprotective activity of fruit samples. Furthermore, studies in maqui and murta with convective hot air drying above 60 °C suggest that the loss of antioxidant activity is compensated by a probable formation of bioactive components directly related to TPC [55,56]. Otherwise, convective and combined convective-infrared conditions (40 °C/800 W) reduced the drying time, resulting in dried samples of murta fruit with the highest TPC [56]. Therefore, convective hot air technology can be a more economical alternative for applying these native fruits, considering their use with the local economy.

Inflammation is one of the newly recognized cardiovascular risk factors resulting from the major consequences of oxidative stress [57]. Therefore, the inhibition of oxidative stress and inflammation could positively influence the formation and development of atherosclerosis. In the present study, the effect on inflammation and oxidative stress of macrophages was also evaluated. The ability of crude peumo extract to quench oxidative radicals in macrophages was five times lower than the ability of pure quercetin. Quercetin is a strong antioxidant capable of removing radicals caused by environmental and toxicological factors [58]. It has previously been demonstrated that quercetin derivatives present in systemic circulation after quercetin consumption can act as a potent antioxidant and anti-inflammatory agent and may contribute to the overall biological activity of the quercetin-rich diet [59]. Besides quercetin, the presence of epicatechin and phytocyanidin was also confirmed in the extract. Phytocyanidin is an even stronger antioxidant than quercetin, and epicatechin has a similar or slightly lower effect than quercetin on serum oxidative stress indicators in D-galactose-treated mice [60]. Our data, provided in Table 5, demonstrate the excellent anti-inflammatory activity of the extract, which had better immunomodulatory effects than the reference compounds quercetin and indomethacin, used as standards. This may indicate that quercetin was not the only anti-inflammatory compound in the extract and that the extract contained other more active compounds or compounds acting synergistically with quercetin. These compounds could be either procyanidins [61] or luteolin [62], and catechins [63]. Another explanation may be some modification of flavonoids that makes them more bioavailable. 

The anti-inflammatory and antioxidant PCAs for a Peumo lyophilized sample (Figure A2) both present an expected behavior. From this analysis, we have an insight that peumo and quercetin behave similarly, with a highly negative correlation with antioxidant capacity analyses (TFC and ORAC); however, TPC was not related with the other variables (orthogonal), when we expected a positive correlation with TFC and ORAC. Possibly, for TPC, this could be a technical interference, such as sugar molecules related to SSC (Table 1) in the 2018 season. On the other hand, the anti-inflammatory effect variables behave opposite between IL-6, TNF-α, and NO.

A positive effect of ripe peumo fruit extracts as anti-inflammatory was found; therefore, the vasoprotective effect of peumo extract was evaluated. The endothelium exerts several vasoprotective effects, such as vasodilation and inhibition of inflammatory responses. Endothelial dysfunction, signaled by impaired endothelium-dependent vasodilation, is an early marker for atherosclerosis [64]. In addition to endothelial dysfunction, inflammation is also associated with increased cardiovascular diseases and subclinical atherosclerosis.

In the present study, the deterioration of endothelium-dependent relaxation, induced by high glucose in the aortic rings, was partially reversed by the extract of ripe peumo fruits (Figure 4), suggesting that this extract may prevent vascular damage caused by high glucose levels. An example is postprandial hyperglycemia, a phenomenon associated with an increased risk of cardiovascular disease in diabetic and non-diabetic subjects [65]. Previous studies indicate that the vasoprotective mechanism depended on bioactive molecules. Therefore, flavonols demonstrated an endothelium-dependent relaxation, while flavones displayed an endothelium-independent behavior. Endothelium removal decreased the sensitivity of responses to flavonols, without affecting the maximal relaxation [66]. It was described that a murta extract [67], rich in catechin, gallic acid, myricetin, quercetin, quercetin-3-β-D-glucoside, and kaempferol, and an arrayan extract [26], containing quercetin-3-rutinoside, malvidin-3-arabinoside, peonidin-3-galactoside, peonidin-3-arabinoside, and petunidin-3-arabinoside, displayed vasodilator activity in aortic rings that was dependent of the extract concentration, with a potential association to endothelium. Data of this paper clearly demonstrate the strong potential of peumo extract in the complex prevention of atherosclerosis. However, additional studies are necessary to determine the relationship between the bioactive compounds of peumo extract and endothelium dependence in these extracts’ vasoprotective effect.

## 5. Conclusions

This study demonstrates important aspects of peumo developmental stages as an interesting source of bioactive compounds. The development of this fruit was described by an increase in length and change in color. Furthermore, the decrease in CO_2_ production during ripening, without ethylene detection during maturation, suggests a non-climacteric behavior for this fruit. Notably, the high antioxidant capacity determined by ORAC and cellular assay and the anti-inflammatory and vasoprotective activity of methanol extracts of ripe peumo could be associated with the presence of different bioactive molecules: quercetins, procyanidins, catechin and epicatechin, chlorogenic acid, and alkaloids, among others, observed by HPLC/MS. These results suggest that peumo fruit could be used as a functional ingredient. The results are promising, considering that peumo is a species of easy domestication and can grow in low irrigation conditions. Therefore, future studies of the effect of postharvest and processing in these native fruits are critical to strengthening their potential use as a new food or ingredient.

## Figures and Tables

**Figure 1 antioxidants-10-01997-f001:**
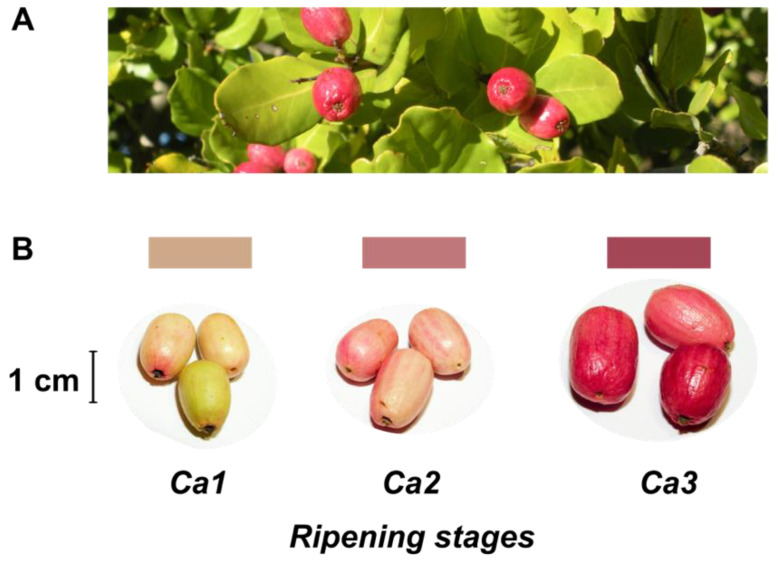
Leaves and fruits of peumo tree. (**A**) Leaves and fruits of peumo [*Cryptocarya alba* (Molina) Looser]; (**B**) Three different development stages of peumo fruit. Photography credit: Lida Fuentes.

**Figure 2 antioxidants-10-01997-f002:**
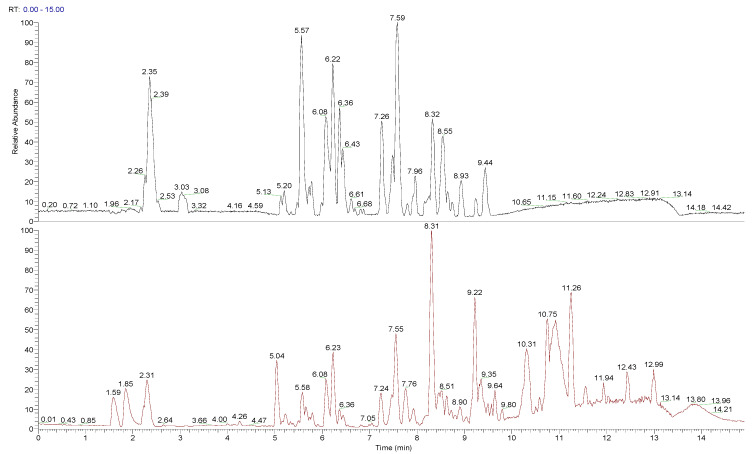
Total ion chromatogram (TIC) for peumo, presented in negative and positive modes.

**Figure 3 antioxidants-10-01997-f003:**
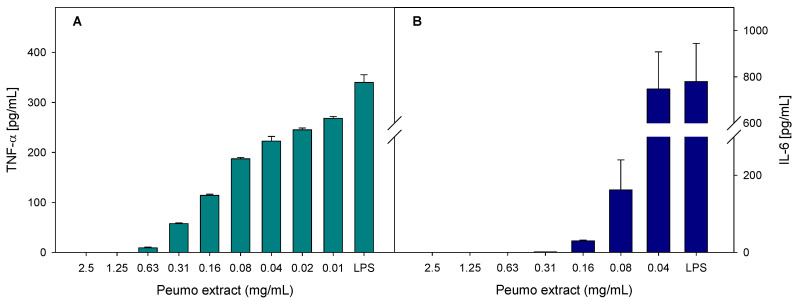
Effect of peumo extract on relative production of TNF-α (**A**) and IL-6 (**B**), by LPS-stimulated RAW 265.4 macrophages. Data represent the average of three repetitions, with a corresponding standard error of the mean.

**Figure 4 antioxidants-10-01997-f004:**
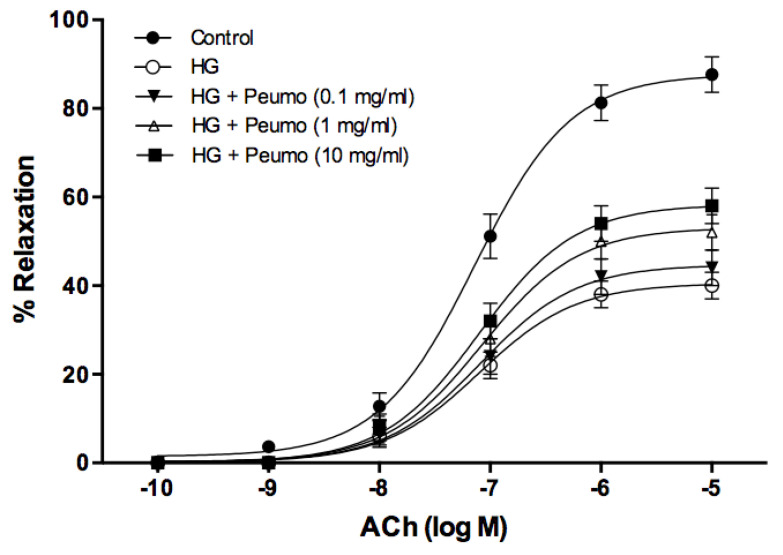
Vasoprotector effect of peumo extract on aortic rings. Concentration-response curves to ACh (0.0001–10 µM) in precontracted rings with 0.1 µM of PE. Before performing the concentration-response curves to ACh, the rings were pre-incubated for 3 h under the following conditions: control, high glucose (5 mM D-glucose) (HG), and in the presence of HG plus extracts of peumo (0.1, 1, and 10 mg/mL) (Ca3 stage). The results (mean ± SE) are expressed as a percentage of relaxation (% relaxation).

**Table 1 antioxidants-10-01997-t001:** Changes in physicochemical and physiological parameters during fruit development of peumo fruit for the 2017 and 2018 harvest seasons.

		Developmental Stages					
Parameter		Harvest Season 2017			Harvest Season 2018		
		Ca1	Ca2	Ca3	Ca1	Ca2	Ca3
Ethylene production		nd*	nd*	nd*	nd*	nd*	nd*
Oxygen consumption (mg Kg^−1^ h^−1^)		6.73 ± 0.00 a	5.85 ± 0.04 b	1.88 ± 0.04 c	6.98 ± 0.01 a	5.99 ± 0.02 b	2,07 ± 0.02 c
CO_2_ production (mg Kg^−1^ h^−1^)		8.53 ± 0.00 a	7.49 ± 0.04 b	3.29 ± 0.04 c	6.86 ± 0.04 a	6.43 ± 0.02 b	5.30 ± 0.51 c
Firmness (N)		7.54 ± 0.20 a	5.99 ± 0.23 b	4.97 ± 0.15 c	7.93 ± 0.76 a	6.04 ± 1.14 b	5.03 ± 0.65 c
pH		5.66 ± 0.00 c	5.75 ± 0.00 b	5.90 ± 0.01 a	6.12 ± 0.05 a	6.26 ± 0.02 a	6.26 ± 0.03 a
TA (%)		nd*	nd*	nd*	nd*	nd*	nd*
SSC (°Brix)		24.72 ± 2.18 b	32.50 ± 1.78 a	38.06 ± 4.16 a	30.00 ± 2.50 a	25.83 ± 1.44 a	29.17 ± 1.44 a
Length (cm)		1.25 ± 0,02 b	1.34 ± 0.02 b	1.59 ± 0.05 a	1.76 ± 0.10 b	2.08 ± 0.04 a	2.10 ± 0.04 a
Diameter (cm)		0.91 ± 0.02 c	0.97 ± 0.01 b	1.08 ± 0.02 a	1.21 ± 0.04 a	1.25 ± 0.02 a	1.22 ± 0.01 a
L/D		1.38 ± 0.02 b	1.38 ± 0.02 b	1.47 ± 0.03 a	1.45 ± 0.05 b	1.66 ± 0.02 a	1.73 ± 0.05 a
FW (g)		0.58 ± 0.03 c	0.79 ± 0.02 b	1.19 ± 0.05 a	1.75 ± 0.17 a	2.08 ± 0.08 a	1.99 ± 0.05 a
DW (g)		0.32 ± 0.02 c	0.46 ± 0.01 b	0.69 ± 0.03 a	1.07 ± 0.11 a	1.26 ± 0.06 a	1.13 ± 0.03 a
Humidity_wet basis_		43.10 ± 0.43 a	41.77 ± 0.54 b	40.34 ± 0.47 b	42.29 ± 0.46 a	41.35 ± 1.20 a	40.70 ± 0.63 b
Humidity_dry basis_		75.76 ± 0.29 a	71.74 ± 0.05 b	67.61 ± 0.07 b	73.27 ± 0.16 a	70.49 ± 0.55 a	66.86 ± 0.12 b
Water activity		0.81 ± 0.01 a	0.78 ± 0.02 a	0.77 ± 0.04 a	0.78 ± 0.02 a	0.76 ± 0.02 a	0.78 ± 0.05 a
Color (CIElab*)	L*	69.67 ± 0.88 a	67.13 ± 1.08 a	68.21 ± 1.20 a	71.83 ± 0.82 a	58.17 ± 1.03 b	43.14 ± 1.10 c
	a*	10.66 ±1.28 b	14.19 ± 1.81 a	13.43 ± 1.89 a	10.64 ± 0.78 c	29.44 ± 1.04 b	41.14 ± 0.76 a
	b*	19.36 ± 0.63 a	20.05 ± 1.38 a	17.80 ± 1.31 b	18.57 ± 0.48 a	11.70 ± 0.59 b	11.10 ± 0.28 b
	C	22.10 ± 1.43 a	24.56 ± 2.28 a	22.31 ± 2.30 a	21.07 ± 2.47 c	31.87 ± 0.98 b	42.61 ± 0.77 a
	h°	61.16 ± 0.56 a	54.71 ± 1.03 b	52.97 ± 1.17 b	60.27 ± 2.30 a	21.92 ± 1.49 b	15.12 ± 0.33 c

Data correspond to the mean ± SE. Different letters point to significant differences between developmental stages in each harvest season (*p* ≤ 0.05). SSC, soluble solids content; TA, titratable acidity; FW, fresh weight; L/D, length and diameter ratio; DW, dry weight. Statistical analysis: One-Way ANOVA. Tukey (*p* < 0.05). nd*, not detected.

**Table 2 antioxidants-10-01997-t002:** Total polyphenol content (TPC), total flavonoid content (TFC), antioxidant capacity by FRAP, TEAC, DPPH, and ORAC methods, during the development of peumo fruits for the 2017 and 2018 harvest seasons. GAE, gallic acid equivalent; QE, quercetin equivalent; TE, trolox equivalent.

Development Stage	TPC [mgGAE/gFW]		TFC [mgQE/gFW]		FRAP [μmol FeSO_4_/gFW]		TEAC [mmol TE/gFW]		DPPH [IC_50_ μg/mL]		ORAC [mmol TE/gFW]	
Season 2017												
Ca1	11.19 ± 0.7.4	b	7.34 ± 0.27	c	28.62 ± 0.95	b	4.34 ± 0.34	c	6.84 ± 0.17	b	n.a.	
Ca2	13.70 ± 1.03	b	8.52 ± 0.33	b	35.96 ± 0.36	a	7.23 ± 0.32	b	6.89 ± 0.54	b	n.a.	
Ca3	17.87 ± 1.57	a	9.21 ± 0.28	a	38.34 ± 0.33	a	8.09 ± 0.22	a	8.72 ± 0.14	a	n.a.	
**Season 2018**												
Ca1	12.85 ± 1.16	c	6.98 ± 0.21	c	29.49 ± 2.36	b	5.02 ± 0.39	b	7.69 ± 0.90	b	0.208 ± 0.010	a
Ca2	15.15 ± 0.71	a	8.46 ± 0.39	b	35.94 ± 1.23	a	7.12 ± 0.18	a	7.09 ± 0.19	a	0.199 ± 0.002	a
Ca3	17.61 ± 0.60	a	9.44 ± 0.18	a	37.08 ± 0.75	a	7.91 ± 0.30	a	8.35 ± 0.53	a	0.188 ± 0.002	a
Blueberry	2.75 ± 0.2		2.12 ± 0.44		4.95 ± 0.28		1.25 ± 0.30		11.36 ± 0.96		0.032 ± 0.000	

Data correspond to the means ±SE of four replicates of fruit mix for each stage and season. Different letters point to significant differences between developmental stages in each parameter (*p* ≤ 0.05). n.a., not analyzed.

**Table 3 antioxidants-10-01997-t003:** Identification of compounds and antioxidant capacity from the methanol extract of peumo fruits, by LC-MS and MS/MS data. The principal peaks were individually analyzed, and the potential molecules were identified. Also, total polyphenol content (TPC), total flavonoid content (TFC), and antioxidant capacity by ORAC methods were determined. GAE, gallic acid equivalent; QE, quercetin equivalent; TE, trolox equivalent.

RT (min)	[M + X]^+^ (m/z)	[M − X]^−^ (m/z)	[M] (m/z)	Fragments	MF	Tentative Compound
2.35	343.1236 [M + H]^+^	341.1112 [M − H]^−^ 683.2303 [2M − H]^−^	342	−89.0227, 101.0226, 119.0330, 143.0329, 161.0433, 179.0537	C_12_H_22_O_11_	Sucrose
5.04	316.2121 [M + H]^+^		315	102.0916, 123.1169, 184.1695, 255.1590	C_19_H_25_NO_3_	Cryprochine Isocryprochine
5.13	867.2132 [M + H]^+^	865.2036 [M − H]^−^	866	−287.0587 −407.0808 −451.1076 −577.1406 −695.1471 713.1580 −739.1739 −847.0959	C_45_H_38_O_18_	Procyanidin C_1_
5.20	579.1500 [M + H]^+^ 1155.2760 [2M + H]^+^	577.1387 [M − H]^−^ 1153.2665 [2M − H]−	578	−289.0744 −407.0809 −425.0918 −451.1076	C_30_H_26_O_12_	Procyanidin B_1_ Procyanidin B_2_
5.57	355.1025 [M + H]^+^ 377.0845 [M + Na]^+^	353.0900 [M − H]^−^ 707.1879 [2M − H]^−^	354	135.0458 179.0362	C_16_H_18_O_9_	4-Caffeoylquinic acid
	579.1500 [M + H]^+^	577.1386 [M − H]^−^	578		C_30_H_26_O_12_	Procyanidin B_1_ Procyanidin B_2_
	867.2131 [M + H]^+^	865.2020 [M − H]^−^	866		C_45_H_38_O_18_	Procyanidin C1
6.08	355.1027 [M + H]^+^ 377.0846 [M + Na]^+^ 731.1894 [2M + Na]^+^	353.0903 [M − H]^−^ 707.1885 [2M − H]^−^	354	191.0575	C_16_H_18_O_9_	Chlorogenic acid
6.22	291.0863 [M + H]^+^ 313.0680 [M + Na]^+^	289.0737 [M − H]^−^ 353.0900 579.1549 [2M − H]^−^	290	−179.0357, 205.0516 245.0811	C_15_H_14_O_6_	Catechin, Epicatechin
6.35	470.1659 [M + H]^+^ 492.1486 [M + Na]^+^	468.1540 [M − H]^−^ 937.3156 [2M − H]^−^	469	292.1217, 424.1651		Unidentified
6.43	355.1024 [M + H]^+^ 731.1821 [2M + Na]^+^	353.0901 [M − H]^−^ 707.1879 [2M − H]^−^	354	−191.0568	C_16_H_18_O_9_	Analogue of chlorogenic acid 4-Caffeoylquinic acid
7.26	465.1031 [M + H]^+^ 487.0849 [M + Na]^+^	463.0912 [M − H]^−^	464	301.0380 178.9999 151.0046	C_21_H_20_O_11_	Isoquercitirin Hyperoside
7.48	435.0926 [M + H]^+^ 457.0743 [M + Na]^+^	433.0801 [M − H]^−^	434	−301.0381	C_20_H_18_O_11_	Reynoutrin Quercetin 3-O-α-D-arabinopyranoside Quercetin 3-O-xyloside
	551.1037 [M + H]^+^ 573.0854 [M + Na]^+^	549.0919	550			(−)-Rubrichalcolactone
		505.1018 [M − H]^−^	504			(6-(5,7-dihydroxy-2-(4-hydroxy-3-methoxyphenyl)-4-oxo-4H-chromen-8-yl)-3,4,5-trihydroxytetrahydro-2H-pyran-2-yl)methyl acetate
7.59	449.1081 [M + H]^+^	447.0963 [M − H]^−^	448	−301.0378	C_21_H_20_O_11_	Quercetin 3-O-α-D-rhamnopyranoside
7.76	353.1361 [M + H]^+^ 376.2120 [M + Na]^+^	351.1412 [M − H]^−^ 375.1473 [M + Na−H]^−^	352	335.1254	C_22_H_40_O_3_	cryptorigidifoliol A
7.96	463.1238 [M + H]^+^	461.1119 [M − H]^−^	462		C_22_H_22_O_11_	Isorhamnetin-3-O-rhamnoside Luteolin 7-O-glucuronide
8.30	360.2155 [M + H]^+^	359.1525	358	313.1465, 327.1466, 341.1624	C_20_H_24_O_6_	(+)-Lariciresinol 4-O-Methylcedrusin
8.93	263.1641 [M + H]^+^ 285.1458 [M + Na]^+^		262	165.0546	C_16_H_22_O_3_	1′R*,3′S*,4′R*,5′S*,6S-6-[(4′-ethyl-9′-oxabicycle[3.3.1]non-6′-en-3′-yl)methyl]- 5,6- dihydro-2H-pyran-2-one
9.21	376.2599 [M + H]^+^ 398.2415 [M + Na]^+^	374.2470 [M − H]^−^	375	209.1284 275.1754 293.1861 302.1864		Unidentified
9.44		315.1620		−118.0428 −163.0409 −271.1726	C_16_H_12_O_7_	Sexangularetin
10.31	305.1747 [M + H]^+^ 327.1568 [M + Na]^+^		304			Unidentified
10.75	247.1694 [M + H]^+^ 269.1513 [M + Na]^+^		246	173.1328 229.1592	C_15_H_18_O_3_	Ethyl 5-hydroxy-7-phenyl-2,6-heptadienoate
11.26	249.1850 [M + H]^+^ 271.1668 [M + Na]^+^		248	133.1016 231.1750	C_16_H_24_O_2_	(4R,6S)-10-Phenyl-1-decene-4,6-diol
Antioxidant capacity of peumo extract						
ORAC (mmol/g DW				0.637 ± 0.061		
TPC (mgGAE/gDW)				23.81 ± 3.06		
TFC (mgQE/gDW)				18.84 ± 3.33		

**Table 4 antioxidants-10-01997-t004:** Concentration of peumo halving the oxygen radicals in cellular antioxidant activity assay (CAA, Raw 264.7 cell line). Data represent the average of three biological repetitions, with a corresponding standard mean error.

Sample	IC_50_ [mg/L]
Peumo	14.8 ± 1.5
Quercetin	6.3 ± 0.6

Data represent the average of three repetitions, with a corresponding standard error of the mean.

**Table 5 antioxidants-10-01997-t005:** The concentration of peumo halves the markers of inflammatory response. Data represent the average of three repetitions, with a corresponding standard error of the mean.

Sample\Activity	IC_50_ [mg/L]		
	NO	TNF-α	IL-6
Peumo	13.2 ± 0.5 b	129.5 ± 3.5 b	40.0 ± 4.0 b
Quercetin	4.4 ± 0.6 a	8.7 ± 0.7 a	5.0 ± 0.5 a
Indomethacin	19.4 ± 0.3 c	>40	>40

Data represent the average of three repetitions, with corresponding standard error of the mean, and was analyzed by one-way ANOVA, followed by a Duncan’s post hoc test. Different letters indicate statistical differences (*p* < 0.05) within one analysis. NO, nitric oxide.

## Data Availability

Data is contained within the article.

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
