# Peer review of "Characterization of Fruit Development, Antioxidant Capacity, and Potential Vasoprotective Action of Peumo (Cryptocarya alba), a Native Fruit of Chile"

_antioxidants, 2021, doi:10.3390/antiox10121997_

Round 1

Reviewer 1 Report

Dear authors,

thank you for this work. It is dedicated to still less-known plant - peumo, and it is clearly seen that you assayed peumo thoroughly. However, before I accept the paper I have still a few remarks and questions:

  1. Latin names and terms (in vitroin vivo) should be in italics in the whole text. Please check it and correct.
  2. Reference positions looked like being set in two-three different styles. Please correct it.
  3. Page 2, line 74: authors compare previous peumo ORAC results with berry fruits. Please provide information about the range how much higher was the ORAC for peumo compared to the berries. 
  4. chapter 2.4 - TPC, TFC and radical scavenging tests: Please provide information on calibration curves used in these tests. Can be r2 values or regression equations. 
  5. page 4, line 145: Why you used blueberry as a reference fruit? The used blueberry fruits were purchased from some grocery store or collected by you? 
  6. chapter 2.4.1 In line 150 you wrote that standard for FRAP is vitamin C, and you give the final resulsts in FeSO4 equivalents. Please delete this confusing part about vitamin C. 
  7. chapter 2.4.2 authors wrote they calculated the TEAC results according to the cited equation. Please write down this equation in your work. Also in line 164 you wrote mM TE/g FW. mM is a concentration of millimole in one liter. Please consider changing that to mmol TE/g FW.
  8. chapter 2.4.3. Please provide equation you used for determination of % inhibition DPPH. It is clear you could not calculate IC50 without this. 
  9. You use two different abbreviations for Trolox equivalents in one study. Once it is TE for ABTS and for ORAC is ET. Please be consequent and use one abbreviation only.
  10. Chapter 2.5. The key problem is that for further tests you applied different method of fruits preparation than in previous chapter. It is clear that properties of the fruits and their chemical composition must be different after the long-term drying in 60°C. Authors repeated only ORAC test for such a changed sample. I found this insufficient. Authors should re-test TPC, TFC ABTS and DPPH to prove that this treatment would not change the properties significantly. 
  11. Chapters 2.5.3 and 2.5.4. You described only the stage of tests in the wellplates. Interesting is how you precultivate the macrophages. After which passage you used the cells? How they were precultivated (the same medium or other) and what was the final confluency? 
  12. In these point you provide the name of the vendor of the reagents and macrophages. Please consider adding this information for all chemicals applied in the work.
  13. Table 2. Please present ORAC results calculated per 1g of FW as you present other results in this Table. It would be more clear.
  14. Table 5. Please do the post-hoc test of significant differences as you did for results presented in previous tables.
  15. Authors should also consider conducting principal component analysis to visualize how different tests relate with each other and how strong they affect the properties of the tested samples. It is worth to consider setting in the one PCA data from Tables 1-2 and in the latter results from Tables 3 (ORAC only), 4 and 5.

Author Response

1. Latin names and terms (in vitro, in vivo) should be in italics in the whole text. Please check it and correct.

As the reviewer suggests, the Latin terms were checked.

2. Reference positions looked like being set in two-three different styles. Please correct it.

As the reviewer suggests, the references were corrected.

3. Page 2, line 74: authors compare previous peumo ORAC results with berry fruits. Please provide information about the range how much higher was the ORAC for peumo compared to the berries.

These suggestions were considered, and ORAC information for native fruits and commercial berries growing in Chile were included in this paragraph.

4. chapter 2.4 - TPC, TFC and radical scavenging tests: Please provide information on calibration curves used in these tests. Can be r2 values or regression equations.

As the reviewer suggests, r2 values and regression equations were included in the method's description.

5. page 4, line 145: Why you used blueberry as a reference fruit? The used blueberry fruits were purchased from some grocery store or collected by you?

The blueberry was used as a reference because is a crop cultivated in the same locality of Peumo.

6. chapter 2.4.1 In line 150 you wrote that standard for FRAP is vitamin C, and you give the final resulsts in FeSO4 equivalents. Please delete this confusing part about vitamin C.

This error was corrected in the text

7. chapter 2.4.2 authors wrote they calculated the TEAC results according to the cited equation. Please write down this equation in your work. Also in line 164 you wrote mM TE/g FW. mM is a concentration of millimole in one liter. Please consider changing that to mmol TE/g FW.

The equation was included, and TEAC values were expressed in mmol TE/g FW

8. chapter 2.4.3. Please provide equation you used for determination of % inhibition DPPH. It is clear you could not calculate IC50 without this.

The equation was included

9. You use two different abbreviations for Trolox equivalents in one study. Once it is TE for ABTS and for ORAC is ET. Please be consequent and use one abbreviation only.

This was corrected

10. Chapter 2.5. The key problem is that for further tests you applied different method of fruits preparation than in previous chapter. It is clear that properties of the fruits and their chemical composition must be different after the long-term drying in 60°C. Authors repeated only ORAC test for such a changed sample. I found this insufficient. Authors should re-test TPC, TFC ABTS and DPPH to prove that this treatment would not change the properties significantly.

As suggested by the reviewer, additional analyses were included for the methanol extract of dry peumo. TheTPC and TFC were re-test in the methanol extract. However, ABTS and DPPH can´t be included due to the short time for the revisions' answers and the lack of some reagents to perform these analyses. Furthermore, we will try to include these last methods in the final version of the manuscript.

11. Chapters 2.5.3 and 2.5.4. You described only the stage of tests in the wellplates. Interesting is how you precultivate the macrophages. After which passage you used the cells? How they were precultivated (the same medium or other) and what was the final confluency?

As the reviewer suggested, all these points were included.

12. In these point you provide the name of the vendor of the reagents and macrophages. Please consider adding this information for all chemicals applied in the work.

As the reviewer suggested, this information was included.

13. Table 2. Please present ORAC results calculated per 1g of FW as you present other results in this Table. It would be more clear.

As the reviewer suggests, the ORAC results were expressed by 1g of FW.

14. Table 5. Please do the post-hoc test of significant differences as you did for results presented in previous tables.

As the reviewer suggests, significant differences were added to Table 5.

15. Authors should also consider conducting principal component analysis to visualize how different tests relate with each other and how strong they affect the properties of the tested samples. It is worth to consider setting in the one PCA data from Tables 1-2 and in the latter results from Tables 3 (ORAC only), 4 and 5.

As the reviewer suggests, PCA and correlation analyses were conducted in the data where these analyzes were feasible to perform. This new information improves the interpretation of our results.

The authors thank the Reviewer comments

Reviewer 2 Report

The research topics are “Characterization of fruit development and potential health benefits of peumo (Cryptocarya alba), a native fruit of Chile". This manuscript is very interesting and the content design is very complete. For all these reasons I recommend minor revisions to the manuscript be considered for publication in this journal.

  1. Table 2 shows that the total polyphenol content (TPC) in the Ca1 – Ca3 maturation period of the two years is higher (about 1.5~1.9 times) than the total flavonoid content (TFC), but the overall discussion focuses on quercetin, especially in Line 473- 550. It is recommended that the author, when reporting on Flavonoids with 3-O-conjugates structure, can appropriately supplement the discussion about polyphenols.

-DOI: 10.1016/j.jep.2020.112559

-DOI: 10.3390/ph13050087

-DOI: 10.1016/j.jnutbio.2017.09.009

-DOI: 10.1016/j.jep.2016.04.040

  1. Conclusions The important findings are not expressed and should be improved. For example, there is no information of U-HPLC/MS.
  2. Should be amended:

-The species names should be the same, Line 93 / Line 318

-Table 2. 11,19 / 11.19

-Literature citation, Line483

-The usage should be consistent, Line 200 / Line 462, Line 354, flavonoids (TFC)

Author Response

1. Table 2 shows that the total polyphenol content (TPC) in the Ca1 – Ca3 maturation period of the two years is higher (about 1.5~1.9 times) than the total flavonoid content (TFC), but the overall discussion focuses on quercetin, especially in Line 473- 550. It is recommended that the author, when reporting on Flavonoids with 3-O-conjugates structure, can appropriately supplement the discussion about polyphenols.

-DOI: 10.1016/j.jep.2020.112559

-DOI: 10.3390/ph13050087

-DOI: 10.1016/j.jnutbio.2017.09.009

-DOI: 10.1016/j.jep.2016.04.040

As the reviewer suggests, the discussion about the polyphenols compounds was improved and the suggested papers were included.

2. Conclusions The important findings are not expressed and should be improved. For example, there is no information of U-HPLC/MS.

As the reviewer suggests, the conclusion was improved, including important information about the bioactive compounds.

3. Should be amended:

-The species names should be the same, Line 93 / Line 318

-Table 2. 11,19 / 11.19

-Literature citation, Line483

-The usage should be consistent, Line 200 / Line 462, Line 354, flavonoids (TFC)

All these suggestions were included.

The authors thank the Reviewer comments

Reviewer 3 Report

The manuscript regards the evaluation of the antioxidant capacity and the tentative description of the chemical composition of peumo fruit (Cryptocarya alba) from central Chile. Moreover, the authors evaluated the anti-inflammatory and vasoprotective activity.

The manuscript is well written and the experimental work is extensive. Anyway, the authors shoud explain the antioxidant results they obtained with the different methods, comparing the results with whose obtained for the blueberry. The ORAC method should be discussed in detail and the authors should explain why no differences were observed in the ORAC method for the second season. The chemical composition was determined only by HPLC-MS analysis but no standard compounds were used. Since some of the compounds identified are commercially available and cheap the authors should verify that the assignement made by MS analysis is confirmed by the retention time with reference compounds.

Author Response

The manuscript is well written and the experimental work is extensive. Anyway, the authors shoud explain the antioxidant results they obtained with the different methods, comparing the results with whose obtained for the blueberry. The ORAC method should be discussed in detail and the authors should explain why no differences were observed in the ORAC method for the second season. The chemical composition was determined only by HPLC-MS analysis but no standard compounds were used. Since some of the compounds identified are commercially available and cheap the authors should verify that the assignement made by MS analysis is confirmed by the retention time with reference compounds.

As the reviewer suggests, the discussion of antioxidant results was improved, and PCA and correlation analysis was done.

In the beginning, our research group (CREAS Lab) did an HPLC-DAD using commercial standards shows catechins, chlorogenic acid and derivates, and quercetins as the major phenolic compounds (data no-showed). However, this technique did not identify several compounds, and a U-HPLC/MS was carried out at Prega University to determine a more significant number of molecules. The compounds were detected in the accordance with recorded MS/MS spectra with online mass spectral libraries (such as ‘METLIN’, ‘mzCloud’) or the scientific literature. For some of the detected compounds, several chromatographic peaks meeting the HRMS criteria were observed, probably indicating the presence of structural isomers. Therefore, the principal bioactive molecules detected by U-HPLC/MS were previously analyzed by standards compounds in HPLC-DAD.

The authors thank the Reviewer comments

Round 2

Reviewer 1 Report

no further remarks.

Congratulations!